## [Transparent Peer Review file · Nature Communications]

Developmental plasticity enables an intestinal tapeworm to adapt to dietary stress

Corresponding Author: Dr Katerina Jirku

Version 0:

Reviewer comments:

Reviewer #1

(Remarks to the Author)

The stated goal of this study is to examine the role of diet in inducing aestivation in the rat tapeworm, *Hymenolepis diminuta*. To do this, the authors performed a series of experiments testing the effects of host diet [Western (W) vs. Accessible fiber (A) diets] on parasite morphology and gene expression as well as the host microbiome, metabolome and immunity. Questions about how host diet might influence interactions between hosts and their parasites are of considerable interest and it is clear from this study that there are complex and intriguing interactions occurring among host, parasite and microbiome that are mediated by diet. However, the experimental design and analyses presented in this study do not adequately disentangle these complexities. Importantly, the study design and results do not appear to support the conclusions that are being drawn, and the findings are not adequately contextualized within the existing literature. More detailed comments are provided below.

Major comments

Introduction

The introduction requires deeper context to introduce the motivating questions and their relevance/significance within the existing literature. For example, it seems relevant to discuss the fact that *H. diminuta* can infect people as part of the motivation for looking at the effects of a Western diet. A brief discussion of the biology of this parasite in humans seems like necessary context for this study.

Even more important, the paper would benefit from a discussion of what is already known about the interplay between host diet, microbiome, metabolome, immunity, and helminth parasitism. This includes information from past studies on *H. diminuta*, which are alluded to on line 81 but never described, as well as from other systems, including other host-helminth systems (e.g. domestic livestock systems).

Finally, the phenomenon of aestivation needs to be better defined and characterized given similar processes known to occur in helminths. Helminths show a range of responses to stress that are similar to aestivation (e.g., arrested/delayed development, hypobiosis, dauer) and some of these processes have been previously linked to host diet. For example, parasitic nematodes can delay their growth or reproduction in response to host diet (e.g., Carvalho et al 2023 <https://doi.org/10.3390/ruminants3040033>). If aestivation is a distinct biological phenomenon from these other well-described helminth stress responses, this should be clearly explained, and the introduction and/or methods should describe how aestivation is being qualitatively or quantitatively identified in this paper.

Methods

It is difficult to understand the experimental design and data analysis without reading the supplementary materials. Importantly, key details on study design and statistical analyses need to be presented in the main text for readers to understand what was done and why. For example, in lines 100-106, the sample size of rats per treatment, dosage of helminths administered, and approach to retrieve and quantify adult worms should be clearly described, along with the number of adult worms retrieved at the endpoint of the experiment for each group. Including the figure from Supplementary Data 2 in the main text would be helpful for visualizing Experiment 2. It is crucial that sample sizes are clearly described in the main text (in the methods/alongside the results) so the power of the approach can be evaluated.

Currently, Experiment 2 is presented as a test of the reversibility of the “aestivation” phenomenon observed in Experiment 1 (lines 106-108), however, it is not clear that Experiments 1 and 2 are testing the same phenomenon. In Experiment 1, *H. diminuta* cysticeroids were administered to rats while on the Western diet or Accessible Fiber diet, and the Western diet appeared to affect their development into adult worms. In Experiment 2, *H. diminuta* cysticeroids were administered to rats while on the Accessible Fiber diet, allowed to develop into adult worms, and then rats were switched to the Western diet. Thus, Experiment 1 appears to test the effects of poor host nutrition on worm maturation from cysticeroids into adults, while Experiment 2 tests how poor host nutrition affects adult worms that have already matured on an accessible fiber host diet. To test the stated question about reversibility of the developmental effects observed in Experiment 1, cysticeroids would need to be administered while rats are fed on a Western diet, then switched to the Accessible Fiber diet. This mismatch between the experiment that was actually done and the claims made needs to be addressed. At present, it does not appear that the authors have directly tested the reversibility of aestivation.

An explanation should be given for why particular diets were chosen and some discussion is needed of the key differences in diet components that were the target of the study and why. A list of ingredients as outlined (lines 92-96) is insufficient.

Results & discussion

The combination of results and discussion into a single section made the study findings and conclusions somewhat hard to follow. One recommendation is to disaggregate these sections. Furthermore, statistics (name of statistical test, sample sizes, test statistics, p-values) should be reported in the main text as support (e.g., Lines 112-116, 163-192, 207-216, 224, 235, 237). Currently, it is difficult to link the statistical methods described in the supplementary material to the results outlined in the main text and there are cases where the statistical methods described in the supplement appear not to entirely explain the described results. There are also cases where incorrect statistical approaches seem to have been used (e.g., Figure 2, Figure 5). It should be clear which statistical methods were used for every result presented and claims should not be made without statistical evidence (e.g. lines 112-117, 147-150).

How was worm colonization determined for the different groups being compared and what does the term “colonized” mean (line 118)? Is a colonized rat an animal that was initially inoculated with worms, that shed eggs, or that had adult worms at the endpoint of the experiment? Depending on the definition of colonization, interpretation of these results may change.

In general, many results have been overinterpreted or inaccurately represented. For example:

Lines 112-117: If the sample size for worm morphology analyses is as low as 2 worms total from each of the 2 diet treatments, as stated in Supplementary Methods (Line 97), then the morphological results described here are not meaningful.

Lines 147-150: Without reporting adult worm counts or mortality rates, it is not possible to draw conclusions about differential worm mortality. Additionally, without mortality data, it is difficult to determine whether the phenomena observed here are due to developmental changes or differences in mortality or establishment between diet treatments.

Lines 211-213: These results do not seem to match the figure caption or reported statistics, which suggest that the made comparison is between diets, not within them. Reporting statistics in the main text would aid in clarity and support.

Lines 250-253: Host immune responses seem like a critical part of the study but are only described briefly. Including a figure with immune data in the full text seems necessary to interpret these statements. All patterns observed in both the Western diet and Accessible Fiber diets should be described. For example, on a Western diet, colonization also had effects on IL4 and IFN-g (Supplementary Data 10) that are not referred to at all in the text. Full, transparent reporting of these results is necessary for interpretation.

Figure 2: This figure truncates some of the time series data presented in Supplementary Figure 3. Presenting the full time series data on egg shedding shown in the supplementary figure and explaining the observed variation between groups while fed the same diet (i.e., days 21-49) would be more transparent. Some of these differences appear significant and are not currently explained in the text. Additionally, the methods or results should explain and show the specific statistical tests used to obtain the significance differences shown on the graph, and explain which specific timepoints, treatments, or individuals are being compared. Currently, the meaning of these significance levels is unclear.

Figure 4: There appears to be inaccuracy in the reporting or visualization of results in this figure. Comparisons in 4A and 4B are unclear and should be indicated on the figure. Visually, figure 4B does not seem to show a statistically different effect of colonization in the A-diet versus the W-diet (the bars on the graph appear identical), although the figure caption reports one. Figures 4C and 4D do not match the findings reported in lines 211-216. These figures also show that each diet is more similar to itself, but do not make within-diet comparisons.

Figure 5: The results reported in the text (Lines 217-221) contradict the statistics presented in this figure caption. While the effect sizes vary, colonization has statistically significant effects on the metabolome for both diets, not just the Western diet. Additionally, to test whether *H. diminuta* alters the small intestinal bacteriome in a diet-dependent manner (lines 530-531) requires testing for an interaction between diet and colonization on these indices, which does not appear to have been done.

Minor comments

Lines 77-80: The relevance of therapeutics for *H. diminuta* is unclear. Negative impacts on humans or animals have not

been described.

Line 114: The statement that worms on the Western diet have a “typical morphology” and “normal length” requires a clear, quantitative description of typical morphology.

Lines 135-139: It's essential to know the average total worm burden in each group at the end point of Experiment 1 in order to interpret this statement. Additionally, it is unclear whether these lines refer to results from Experiment 1 or Experiment 2.

Lines 144-145: The conclusion that “tapeworms markedly reduce their metabolic requirements” in response to a Western diet requires evidence not currently presented in the text.

Lines 288-299: This is the first time that *H. diminuta* is described as a human pathogen. This comes too late in the manuscript to be a convincing broader extension of the work.

Figure 3: This figure is difficult to interpret in its current form; it should highlight specific patterns or genes of interest for greater clarity.

Line 243-247: Fructose is the key factor driving separation between diet groups (according to line 226). Yet in line 245, changes in the metabolome are discussed in the context of glucose, sucrose, and maltose. Providing some interpretation of the key fructose result would be helpful.

Reviewer #2

(Remarks to the Author)

Reviewer #3

(Remarks to the Author)

The authors present a well-designed study investigating how two distinct diets, one fiber-rich and the other representative of a Western dietary pattern, affect the development of the intestinal tapeworm *Hymenolepis diminuta* in a rat model. Their data clearly show that the Western-like diet slows down development of this organism in a reversible manner. They hypothesize that the parasite gets into an aestivation state due to factors related to the Western diet and the influence of this diet on the microbiota. They support their hypothesis with gene expression and metabolomics data. These data are insightful. The main finding is compelling and counterintuitive: not only does it suggest a hidden link between Western lifestyle and reduced tapeworm infections, but it also raises the possibility that tapeworms may evade detection in modern societies due to their underdeveloped forms.

The limitation of the study is that, beyond Figure 2, the analyses become relatively superficial. The causal connections between tapeworm physiology and dietary components (or diet-derived metabolites) are not definitively established and appear to fall outside the current scope of the paper. Furthermore, the study does not address how variations in diet composition might alter the outcome. For instance, how would a modified Western diet formulated to reduce fructose levels in the gut impact the observed effects? Fructose emerges as a potentially important metabolite in the analysis, yet its role remains speculative without experimental testing. Whether the study meets the threshold for publication in this journal without such deeper mechanistic or dietary exploration is ultimately an editorial decision.

Minor comments:

Page 3, lines 93-97: Please clarify what the reported percentages refer to (mass percentage, caloric contribution, etc.)

Page 7, lines 213-216: The conclusion that dietary fiber absence leads to “destabilization” of the microbiota is not fully supported by Figure 4. Please elaborate on how this conclusion is derived from the data.

Page 7, lines 237-239: The claim that specific diacylglycerols and amino acids are present only in animals with *H. diminuta* is not directly supported by the cited figures. Please provide metabolite-level information or clarify the connection.

Version 1:

Reviewer comments:

Reviewer #1

(Remarks to the Author)

Major comments

Introduction and discussion:

We appreciate the additional context the revised manuscript provides in the introduction about diet-microbiome relationships, but the existing literature on helminth development and helminth-diet interactions, particularly as directly related to this system, are still poorly represented. For example, lines 76-78 state that little is known about the effects of host diet on helminths. This is not factual, there is a substantial amount of work linking diet with helminth infections. Therefore, more nuance is needed here to understand the precise gap this study is addressing. In particular, line 404 of the discussion refers to a prior study in this system that found similar effects of diet on *H. diminuta* development as reported in this study, seemingly contradicting statements throughout the paper that little is known about how diet affects helminths (lines 76-78), or whether helminths are capable of developmental plasticity (lines 279-280). In the context of helminth development, contrary to the authors' claims on lines 279-280, it is well-known that parasitic helminths can show flexibility in their developmental strategies (including hypobiosis and similar phenomena) and the authors need to clarify how their study builds on what is already known. The strategy of moving all information about helminth development to the discussion (e.g., lines 272-278) only further detracts from having a transparent discussion of the literature this study is building upon up front in the introduction. Crucially, this information cannot be decoupled from the gap the authors are suggesting exists in the helminth literature.

Methods:

Overall, the methods are well laid-out and much clearer in the revised document. We appreciate the additional detail the authors provide about sample sizes and statistical analyses, and the new visualization of the experimental design (Figure 1). However, we still have some concerns about the authors' presentation of their study design.

First, we agree with the authors that acknowledging experiment 1 and experiment 2 test distinct phenomena is a good approach (that is, experiment 1 tests the effects of a Western diet on development to the adult stage while experiment 2 tests the reversibility of this diet's effects on reproduction when temporarily presented to adult worms). However, while the authors do well at acknowledging these two distinct phenomena in the discussion (lines 293-299), they need to be upfront about this distinction in the methods and results section as well, rather than stating that this second experiment is a test of developmental reversibility, which is misleading (e.g., lines 142-144).

Second, while we appreciate the clarification of how colonization was determined (lines 137-142), these are inconsistent methods for determine colonization depending on the treatment group (egg shedding vs adult worm count at necropsy), which could impact the interpretation of results. The authors need to candidly discuss how this inconsistency in measuring colonization may affect their interpretations.

Results:

We continue to have concerns about the alignment between results presented in some figures and results reported in the text. For example, the Simpson's diversity result presented on lines 212-214 is not supported by Figure 6B, and the figure caption does not contain any additional information to clarify this discrepancy. Furthermore, while the immune data we requested is now discussed in the main text, based on the figure these results are not being represented accurately. For example, Figure 9 shows significant differences between colonized and non-colonized rats for IL-1beta, IFN γ , and IL4, but the text states that there is no differences in any of these cytokines between the groups and goes on to interpret the results based on a seeming erroneous conclusion (lines 253-259). It is not clear to us whether the statistical results/figures are incorrectly reported, whether the authors misinterpreted the statistics in the text, or some other issue. However, this is a major issue that must be addressed.

Minor comments

Line 66: Helminths have not "largely vanished from modern populations". Approximately 2 billion people are infected with helminths globally. If this statement is meant to refer only to Western nations, the authors should state this clearly.

Line 122: A quantitative definition/source for "normal length" in *H. diminuta* is needed here.

Reviewer #2

(Remarks to the Author)

Reviewer #3

(Remarks to the Author)

The authors have addressed my previous minor concerns, and their major revision has significantly clarified the scope of the manuscript. The reorganization of the manuscript, including the revised introduction and the separation of the discussion section, has also improved overall clarity. I have no additional general comments beyond what I stated in my initial review.

Minor comments:

* In the revised and relocated Table 1, the main components of the Western diet are not fully represented. Only butter fat and

cholesterol are listed, whereas the original supplementary table included several additional ingredients. This appears to be a mistake.

* The new discussion is helpful and more clearly framed, but it is quite long. I recommend trimming redundant or lower-priority content to improve focus and readability.

BIOLOGY CENTRE CAS

address: Branišovská 1160/31, 370 05 České Budějovice, Czech Republic

IBAN: CZ39 0300 0000 0006 0077 3445 | SWIFT CODE: CEKOCZPP | VAT No.: CZ60077344

phone: +420 387 771 111 (telephone exchange) | www.bc.cas.cz | e-mail: bc@bc.cas.cz

RESPONSES TO REVIEWERS` COMMENTS

REVIEWERS #1 & #2:

R1 & 2 comment: The stated goal of this study is to examine the role of diet in inducing aestivation in the rat tapeworm, *Hymenolepis diminuta*. To do this, the authors performed a series of experiments testing the effects of host diet [Western (W) vs. Accessible fiber (A) diets] on parasite morphology and gene expression as well as the host microbiome, metabolome and immunity. Questions about how host diet might influence interactions between hosts and their parasites are of considerable interest and it is clear from this study that there are complex and intriguing interactions occurring among host, parasite and microbiome that are mediated by diet. However, the experimental design and analyses presented in this study do not adequately disentangle these complexities. Importantly, the study design and results do not appear to support the conclusions that are being drawn, and the findings are not adequately contextualized within the existing literature. More detailed comments are provided below.

A: We thank the reviewers for their constructive and thoughtful comments. We believe that some of the initial concerns may have arisen from the fact that our manuscript was directly transferred from the original submission to *Nature*, where strict formatting limitations required us to condense the main text and move several key sections to the Supplementary Materials. In the revised version, we have reintegrated these important details into the main text and substantially expanded the contextualization of our findings within the existing literature. We have also clarified how the experimental design and analyses support the conclusions, thereby addressing the reviewers' concerns regarding the interpretation of host–helminth–microbiome interactions.

R1 & 2 comment: Introduction

Authors` note – see response for this section below.

The introduction requires deeper context to introduce the motivating questions and their relevance/significance within the existing literature. For example, it seems relevant to discuss the fact that *H. diminuta* can infect people as part of the motivation for looking at the effects of a Western diet. A brief discussion of the biology of this parasite in humans seems like necessary context for this study.

Even more important, the paper would benefit from a discussion of what is already known about the interplay between host diet, microbiome, metabolome, immunity, and helminth parasitism. This includes information from past studies on *H. diminuta*, which are alluded to on line 81 but never described, as well as from other systems, including other host-helminth systems (e.g. domestic livestock systems).

Finally, the phenomenon of aestivation needs to be better defined and characterized given similar processes known to occur in helminths. Helminths show a range of responses to stress that are similar to aestivation (e.g., arrested/delayed development, hypobiosis, dauer) and some of these processes have been previously linked to host diet. For example, parasitic nematodes can delay their growth or

reproduction in response to host diet (e.g., Carvalho et al 2023 <https://doi.org/10.3390/ruminants3040033>). If aestivation is a distinct biological phenomenon from these other well-described helminth stress responses, this should be clearly explained, and the introduction and/or methods should describe how aestivation is being qualitatively or quantitatively identified in this paper.

A: We thank the reviewers for this valuable suggestion. The introduction has been fully revised to provide a comprehensive overview of the biology of *Hymenolepis diminuta*, its current relevance in the context of helminth therapy, and the known interactions among host diet, microbiome, metabolome, immunity, and helminth infections (lines 60–114). In addition, references describing studies on *H. diminuta* as well as relevant findings from other host–helminth systems have been incorporated.

The terminology and conceptual distinctions between aestivation and other stress responses in helminths, such as hypobiosis, dauer, arrested development etc are now addressed in the opening part of the Discussion, where they are more logically placed and compared directly with the relevant literature on model helminths (lines 275–351).

R1 & 2 comment: Methods

It is difficult to understand the experimental design and data analysis without reading the supplementary materials. Importantly, key details on study design and statistical analyses need to be presented in the main text for readers to understand what was done and why. For example, in lines 100-106, the sample size of rats per treatment, dosage of helminths administered, and approach to retrieve and quantify adult worms should be clearly described, along with the number of adult worms retrieved at the endpoint of the experiment for each group. Including the figure from Supplementary Data 2 in the main text would be helpful for visualizing Experiment 2. It is crucial that sample sizes are clearly described in the main text (in the methods/alongside the results) so the power of the approach can be evaluated.

A: We fully agree with this comment. This discrepancy arose because the manuscript was originally prepared for submission to *Nature*, where formatting constraints led to a condensed Methods section and reliance on Supplementary Materials. Here, in the revised version, the text has been thoroughly adapted to meet the guidelines of *Nature Communications*. All requested information, including sample sizes, helminth dosage, and methods for worm retrieval and quantification, has been incorporated into the main text (see lines 459-523; or full M&M section 459-681), and the designs of both experiments are now clearly visualized in the revised Figure 1.

R1 & 2 comment: Currently, Experiment 2 is presented as a test of the reversibility of the “aestivation” phenomenon observed in Experiment 1 (lines 106-108), however, it is not clear that Experiments 1 and 2 are testing the same phenomenon. In Experiment 1, *H. diminuta* cysticeroids were administered to rats while on the Western diet or Accessible Fiber diet, and the Western diet appeared to affect their

development into adult worms. In Experiment 2, *H. diminuta* cysticeroids were administered to rats while on the Accessible Fiber diet, allowed to develop into adult worms, and then rats were switched to the Western diet. Thus, Experiment 1 appears to test the effects of poor host nutrition on worm maturation from cysticeroids into adults, while Experiment 2 tests how poor host nutrition affects adult worms that have already matured on an accessible fiber host diet. To test the stated question about reversibility of the developmental effects observed in Experiment 1, cysticeroids would need to be administered while rats are fed on a Western diet, then switched to the Accessible Fiber diet. This mismatch between the experiment that was actually done and the claims made needs to be addressed. At present, it does not appear that the authors have directly tested the reversibility of aestivation.

A: We sincerely thank the reviewers for this insightful and constructive comment, which we consider key for refining and strengthening the manuscript. We fully agree with the concern and, during the revision, carefully considered two possible ways to address it:

Option 1 (implemented in this revision): In the revised manuscript, we have substantially strengthened the conceptual and textual integration of both experiments. We placed particular emphasis on clarifying that they describe two distinct yet related phenomena occurring in *H. diminuta* under Western diet conditions, depending on the developmental stage of the helminth at which dietary stress occurs (see lines in sections Results – 119-1171 and Discussion – 275-310). With this revision, we have eliminated the previous ambiguity, and the manuscript now clearly explains that Experiment 1 and Experiment 2 were designed to test two different processes, directly addressing the reviewer’s concern about a mismatch between the experimental design and the conclusions. We believe this change has considerably improved the clarity and logic of the manuscript. At the same time, it opens an underexplored area in helminths with therapeutic potential, helping to explain discrepancies in clinical outcomes of helminth therapy and highlighting the importance of host diet in such applications. We are confident that this solution best addresses the reviewers’ concerns and provides the manuscript with greater clarity and broader relevance.

Option 2 (future direction): Alternatively, one could design a new experiment specifically to link the two current experiments, particularly to directly test the reversibility of “aestivation” in juvenile tapeworms colonizing rats under Western diet conditions. While such an experiment would be theoretically valuable, in practice it would represent a highly complex, long-term, and high-risk project. The experiment itself would require at least 2.5 months, preparation another 1.5 months, and several further months for sample processing and data analysis. Moreover, because the colonization success rate of rats on the Western diet is relatively low (~50%), the experiment would likely need to be repeated, extending the overall timeline to more than one year. For transparency, we provide the outline of such a potential design below this paragraph (Design 1), which illustrates both the conceptual value and the practical limitations of this approach. Given these constraints, we consider this option beyond the feasible scope of the present study and instead regard it as an important avenue for future, dedicated research aimed at a deeper understanding of *H. diminuta* physiology under different dietary regimes and its immunomodulatory effects on the host.

Design 1: Schematic outline of a potential experimental setup aimed at directly testing the reversibility of *Hymenolepis diminuta* aestivation under Western diet conditions. The design combines elements of Experiments 1 and 2, including colonization of rats with cysticercoids under Western diet, dietary transitions to Accessible Fiber diet, and parallel monitoring of worm development, morphology, and egg production. This conceptual scheme is provided for transparency to illustrate the value of such an approach as well as its practical limitations.

R1 & 2 comment: An explanation should be given for why particular diets were chosen and some discussion is needed of the key differences in diet components that were the target of the study and why. A list of ingredients as outlined (lines 92-96) is insufficient.

A: We agree that the rationale for diet selection should be described in greater detail in the manuscript. Our aim was to create a strong nutritional contrast between two extremes, a highly processed Western diet and a natural, hunter gatherer-like diet. This allowed us to capture clear differences in host and helminth responses in a model that dramatically reflects widespread cultural changes. As the reviewer points out, this approach does not allow us to determine the effects of specific components in the diet. However, the results based on this approach are clinically relevant.

In the revised version, we have therefore expanded the explanation - aspects of the modern Western diet are now described in the Introduction (lines 81-89), and the rationale for diet selection is discussed in more detail both at the end of the Introduction (lines 106-114) and at the beginning of the Methods section (lines 460-464). In addition, the table summarizing the composition of both diets has been moved from the Supplementary Materials into the main text (Table 1) and further elaborated to highlight the key components underlying the dietary contrast examined in this study.

R1 & 2 comment: Results & discussion

The combination of results and discussion into a single section made the study findings and conclusions somewhat hard to follow. One recommendation is to disaggregate these sections.

A: We fully agree with this comment. The combined Results and Discussion section was a relic of the original *Nature* submission format. In the revised manuscript, these sections have been separated and

thoroughly revised in accordance with *Nature Communications* guidelines, which we believe has considerably improved the clarity and readability of the study.

R1 & 2 comment: Furthermore, statistics (name of statistical test, sample sizes, test statistics, p-values) should be reported in the main text as support (e.g., Lines 112-116, 163-192, 207-216, 224, 235, 237). Currently, it is difficult to link the statistical methods described in the supplementary material to the results outlined in the main text and there are cases where the statistical methods described in the supplement appear not to entirely explain the described results. There are also cases where incorrect statistical approaches seem to have been used (e.g., Figure 2, Figure 5). It should be clear which statistical methods were used for every result presented and claims should not be made without statistical evidence (e.g. lines 112-117, 147-150).

A: We fully agree with the reviewers on this important point. In the revised manuscript, the Results section (lines 118-271) has been thoroughly revised and all statistical details are now reported directly in the main text, including the names of the tests used, sample sizes, test statistics, and p-values. This ensures that each result is clearly supported by the corresponding statistical evidence.

In addition, the statistical analysis for egg counts (previously Figure 2, now Figure 3) has been carefully re-evaluated and revised. The updated methodology is described in the Methods section (lines 502-523), the revised results are presented in the Results section (lines 144-171), and the corrected data are shown in Figure 3.

R1 & 2 comment: How was worm colonization determined for the different groups being compared and what does the term “colonized” mean (line 118)? Is a colonized rat an animal that was initially inoculated with worms, that shed eggs, or that had adult worms at the endpoint of the experiment? Depending on the definition of colonization, interpretation of these results may change.

A: We thank the reviewer for this important comment, which indeed requires clearer explanation in the main text.

In this study, we define “colonization” as the presence of *H. diminuta* in the host gastrointestinal tract in either adult or developmentally arrested form, irrespective of reproductive activity. The method of confirming colonization, however, differed between diet groups due to diet-dependent differences in helminth development. In rats maintained on the Accessible Fiber diet, colonization was confirmed by positive egg detection using Sheather’s flotation method after the prepatent period. In rats maintained on the Western diet, no eggs were detected after the prepatent period, but endpoint necropsies consistently revealed stunted worms, confirming helminth presence despite the absence of reproductive maturity. Thus, the definition of colonization was consistent across groups, while the method of confirmation differed depending on diet. This clarification has now been included in the revised Results section (lines 137–142).

R1 & 2 comment: In general, many results have been overinterpreted or inaccurately represented. For example:

Lines 112-117: If the sample size for worm morphology analyses is as low as 2 worms total from each of the 2 diet treatments, as stated in Supplementary Methods (Line 97), then the morphological results described here are not meaningful.

A: We thank the reviewer for this very helpful comment as usual. The statement in the M&M section was an unfortunate misrepresentation. In fact, the total number of adult worms analyzed was 16 from the Accessible Fiber diet group (2 worms per rat from 8 rats with 100% colonization success) and 8 from the Western diet group (up to 2 worms per colonized rat, with approximately 50% colonization success). This has now been corrected and clarified in the Methods section (lines 559–562).

R1 & 2 comment: Lines 147-150: Without reporting adult worm counts or mortality rates, it is not possible to draw conclusions about differential worm mortality. Additionally, without mortality data, it is difficult to determine whether the phenomena observed here are due to developmental changes or differences in mortality or establishment between diet treatments.

A: We fully agree with the concern. In the revised manuscript, statements referring to worm mortality have been removed and the relevant sections have been carefully rephrased to avoid overinterpretation (see lines 120–142). The text now clearly distinguishes between reduced colonization success and developmental arrest under the Western diet.

R1 & 2 comment: Lines 250-253: Host immune responses seem like a critical part of the study but are only described briefly. Including a figure with immune data in the full text seems necessary to interpret these statements. All patterns observed in both the Western diet and Accessible Fiber diets should be described. For example, on a Western diet, colonization also had effects on IL4 and IFN-g (Supplementary Data 10) that are not referred to at all in the text. Full, transparent reporting of these results is necessary for interpretation.

A: In the revised manuscript, the immunological data have been incorporated directly into the main text, together with a dedicated figure (Figure 9). All relevant results are now systematically reported in the Results section (lines 255–271), described in more detail in the Methods (lines 669-681), and further discussed in the Discussion (lines 411–433) as well as in Figure 9. This ensures full transparency and improves the interpretability of host immune responses under both dietary conditions.

R1 & 2 comment: Figure 2: This figure truncates some of the time series data presented in Supplementary Figure 3. Presenting the full time series data on egg shedding shown in the supplementary figure and explaining the observed variation between groups while fed the same diet (i.e., days 21-49) would be more transparent. Some of these differences appear significant and are not currently explained in the text. Additionally, the methods or results should explain and show the specific statistical tests used to obtain the significance differences shown on the graph, and explain which specific timepoints, treatments, or individuals are being compared. Currently, the meaning of these significance levels is unclear.

A: The text of the revised manuscript has been revised as suggested. The full time series data previously shown in Supplementary Figure 3 have been incorporated into the main text (now Figure 3). The results are now accompanied by a detailed description, including the statistical tests applied, explicit labeling of the comparisons (groups and time points), and appropriate interpretation of the observed variation. This revision ensures full transparency and clarifies the meaning of the significance levels displayed in the figure.

Authors' note – see explanation for these 3 comments below:

R1 & 2 comment: Lines 211-213: These results do not seem to match the figure caption or reported statistics, which suggest that the made comparison is between diets, not within them. Reporting statistics in the main text would aid in clarity and support.

Figure 4: There appears to be inaccuracy in the reporting or visualization of results in this figure. Comparisons in 4A and 4B are unclear and should be indicated on the figure. Visually, figure 4B does not seem to show a statistically different effect of colonization in the A-diet versus the W-diet (the bars on the graph appear identical), although the figure caption reports one. Figures 4C and 4D do not match the findings reported in lines 211-216. These figures also show that each diet is more similar to itself, but do not make within-diet comparisons.

Figure 5: The results reported in the text (Lines 217-221) contradict the statistics presented in this figure caption. While the effect sizes vary, colonization has statistically significant effects on the metabolome for both diets, not just the Western diet. Additionally, to test whether *H. diminuta* alters the small intestinal bacteriome in a diet-dependent manner (lines 530-531) requires testing for an interaction between diet and colonization on these indices, which does not appear to have been done.

A: We thank the reviewer for their detailed feedback regarding the results and visualization in Figures 4 (now Figure 6) and 5 (now Figure 7). The inconsistencies arose during the final stages of manuscript preparation, when results were recalculated and additional analyses included shortly before submission, while the corresponding text and Methods were not fully updated. These discrepancies have now been corrected in the major revision (see Results, lines 199–233, and Methods, lines 585–619). All statistical tests are now clearly reported in the main text, with explicit indication of the compared groups in both the text and figure legends. Ambiguities related to both figures have been clarified in the Results sections and figure legends, and the interpretation of the comparisons is now explicitly stated.

R1 & 2 comment: Minor comments

- Lines 77-80: The relevance of therapeutics for *H. diminuta* is unclear. Negative impacts on humans or animals have not been described.
- Line 114: The statement that worms on the Western diet have a “typical morphology” and “normal length” requires a clear, quantitative description of typical morphology.
- Lines 135-139: It’s essential to know the average total worm burden in each group at the end point of Experiment 1 in order to interpret this statement. Additionally, it is unclear whether these lines refer to results from Experiment 1 or Experiment 2.
- Lines 144-145: The conclusion that “tapeworms markedly reduce their metabolic requirements” in response to a Western diet requires evidence not currently presented in the text.
- Lines 288-299: This is the first time that *H. diminuta* is described as a human pathogen. This comes too late in the manuscript to be a convincing broader extension of the work.
- Figure 3: This figure is difficult to interpret in its current form; it should highlight specific patterns or genes of interest for greater clarity.
- Line 243-247: Fructose is the key factor driving separation between diet groups (according to line 226). Yet in line 245, changes in the metabolome are discussed in the context of glucose, sucrose, and maltose. Providing some interpretation of the key fructose result would be helpful.

A: All minor comments have been accepted and the manuscript revised accordingly.

REVIEWER #3: COMMENTS AND SUGGESTIONS FOR AUTHORS

R3 comment: The authors present a well-designed study investigating how two distinct diets, one fiber-rich and the other representative of a Western dietary pattern, affect the development of the intestinal tapeworm *Hymenolepis diminuta* in a rat model. Their data clearly show that the Western-like diet slows down development of this organism in a reversible manner. They hypothesize that the parasite gets into an aestivation state due to factors related to the Western diet and the influence of this diet on the microbiota. They support their hypothesis with gene expression and metabolomics data. These data are insightful. The main finding is compelling and counterintuitive: not only does it suggest a hidden link between Western lifestyle and reduced tapeworm infections, but it also raises the possibility that tapeworms may evade detection in modern societies due to their underdeveloped forms.

A: We thank the reviewer for their thoughtful summary and encouraging feedback. We greatly appreciate the recognition of our study design and findings, and we are pleased that the broader implications of our work were clearly highlighted. This positive assessment is highly motivating for our future research.

R3 comment: The limitation of the study is that, beyond Figure 2, the analyses become relatively superficial. The causal connections between tapeworm physiology and dietary components (or diet-derived metabolites) are not definitively established and appear to fall outside the current scope of the paper. Furthermore, the study does not address how variations in diet composition might alter the outcome. For instance, how would a modified Western diet formulated to reduce fructose levels in the gut impact the observed effects? Fructose emerges as a potentially important metabolite in the analysis, yet its role remains speculative without experimental testing. Whether the study meets the threshold for publication in this journal without such deeper mechanistic or dietary exploration is ultimately an editorial decision.

A: We thank the reviewer for this insightful and relevant comment, which rightly touches on the current scope of our study. To our knowledge, this is among the first studies to investigate the physiological response of *H. diminuta* to two contrasting host diets and to highlight the emergence of a previously unrecognized diet-associated developmental state. We see this work as the beginning of a new research direction in helminth biology, which will need to be developed further in future studies and, as our findings suggest, will not be straightforward.

The study was not designed to dissect causal mechanisms of individual dietary components, which would require a series of additional experiments—constituting a separate, more complex project. We agree, however, that fructose emerges from our data as a potentially interesting metabolite. Its role remains speculative at this stage, as it has not been experimentally tested. To reflect this important point, we have revised the Discussion (lines 395-402) and Conclusions (lines 454-456) to explicitly acknowledge fructose as a candidate metabolite and to emphasize that targeted experiments focusing on fructose and other diet-derived metabolites represent a key direction for follow-up research.

R3 comment: Minor comments:

- Page 3, lines 93-97: Please clarify what the reported percentages refer to (mass percentage, caloric contribution, etc.) [ad 1 response below]
- Page 7, lines 213-216: The conclusion that dietary fiber absence leads to "destabilization" of the microbiota is not fully supported by Figure 4. Please elaborate on how this conclusion is derived from the data. [ad 2 response below]
- Page 7, lines 237-239: The claim that specific diacylglycerols and amino acids are present only in animals with *H. diminuta* is not directly supported by the cited figures. Please provide metabolite-level information or clarify the connection. [ad 3 response below]

A: We agree with the reviewer's comments and have addressed all three points in the major revision.

1. The composition of both diets has been corrected and clarified in Table 1, with values now explicitly expressed as mass percentage of diet (% w/w) or per kg diet.
2. The description of microbiota results has been revised in the Results and Discussion sections (lines 199–233 and 353–384), where we now report reduced microbial richness and altered community structure instead of using the less precise term "destabilization."
3. The metabolomic results have been clarified, with absolute metabolite-level data provided in the dedicated Figshare repository (<https://figshare.com/s/df77505840944a777748>). In the text, we now describe these metabolites as "consistently enriched in colonized animals" rather than "present only."

RESPONSES TO REVIEWERS' COMMENTS

REVIEWER #1 & #2: (REMARKS TO THE AUTHOR):

Major comments

Introduction and discussion:

We appreciate the additional context the revised manuscript provides in the introduction about diet-microbiome relationships, but the existing literature on helminth development and helminth-diet interactions, particularly as directly related to this system, are still poorly represented. For example, lines 76-78 state that little is known about the effects of host diet on helminths. This is not factual, there is a substantial amount of work linking diet with helminth infections. Therefore, more nuance is needed here to understand the precise gap this study is addressing. In particular, line 404 of the discussion refers to a prior study in this system that found similar effects of diet on *H. diminuta* development as reported in this study, seemingly contradicting statements throughout the paper that little is known about how diet affects helminths (lines 76-78), or whether helminths are capable of developmental plasticity (lines 279-280). In the context of helminth development, contrary to the authors' claims on lines 279-280, it is well-known that parasitic helminths can show flexibility in their developmental strategies (including hypobiosis and similar phenomena) and the authors need to clarify how their study builds on what is already known. The strategy of moving all information about helminth development to the discussion (e.g., lines 272-278) only further detracts from having a transparent discussion of the literature this study is building upon up front in the introduction. Crucially, this information cannot be decoupled from the gap the authors are suggesting exists in the helminth literature.

A: We thank the reviewer for this very constructive comment, which indeed helped us refine the conceptual coherence of the manuscript. We agree that our previous wording understated the existing body of work on helminth developmental plasticity and helminth–diet interactions. At the same time, we are constrained by the journal's limits (1000 words for the Introduction, 5000 words for the full manuscript, and a maximum of 70 references). The current version is already close to all three limits, and due to the multidisciplinary scope of the study we have exceeded the reference limit; we are now actively consolidating citations where possible.

In response to the reviewer's concerns, we have substantially revised the relevant sections to more accurately reflect the state of the literature:

- The original statement suggesting that “*little is known about dietary effects on helminths...*” (former lines 76–78) has been rewritten to acknowledge existing studies across different host–parasite systems, including diet-dependent shifts in helminth persistence and developmental arrest (now lines 76–82).
- We improved the contextual link between historical observations of nutritional sensitivity in *Hymenolepis diminuta* and the rationale for our study design (lines 107-108).

BIOLOGY CENTRE CAS

address: Branišovská 1160/31, 370 05 České Budějovice, Czech Republic

IBAN: CZ39 0300 0000 0006 0077 3445 | SWIFT CODE: CEKOCZPP | VAT No.: CZ60077344

phone: +420 387 771 111 (telephone exchange) | www.bc.cas.cz | e-mail: bc@bc.cas.cz

- We incorporated the relevant reference (Myhill et al., 2018, 2020; Carvalho et al., 2023).
- We further revised the passages in the Discussion referring to helminth developmental flexibility (former lines 279–280), clarifying that plasticity — including hypobiosis — is already established in some helminth taxa and situating our findings within this broader framework (now lines 282–284).

These revisions address the reviewer's concerns while remaining within the journal's strict word and reference limits. We believe the updated text now provides a clearer and more accurate representation of the existing literature and how our study builds upon it.

Methods:

Overall, the methods are well laid-out and much clearer in the revised document. We appreciate the additional detail the authors provide about sample sizes and statistical analyses, and the new visualization of the experimental design (Figure 1). However, we still have some concerns about the authors' presentation of their study design.

First, we agree with the authors that acknowledging experiment 1 and experiment 2 test distinct phenomena is a good approach (that is, experiment 1 tests the effects of a Western diet on development to the adult stage while experiment 2 tests the reversibility of this diet's effects on reproduction when temporarily presented to adult worms). However, while the authors do well at acknowledging these two distinct phenomena in the discussion (lines 293-299), they need to be upfront about this distinction in the methods and results section as well, rather than stating that this second experiment is a test of developmental reversibility, which is misleading (e.g., lines 142-144).

A: We agree with the reviewer's point and have revised the Methods and Results sections to clearly distinguish between the two experimental aims (Results: lines 123–125, 146–149; Methods: lines 505–506, 521–522).

Second, while we appreciate the clarification of how colonization was determined (lines 137-142), these are inconsistent methods for determine colonization depending on the treatment group (egg shedding vs adult worm count at necropsy), which could impact the interpretation of results. The authors need to candidly discuss how this inconsistency in measuring colonization may affect their interpretations.

A: We thank the reviewer for this important point. We have clarified the methodological differences in the Results (lines 133–139) and added a concise discussion of how egg-based detection versus necropsy confirmation may influence interpretation (Discussion, lines 315–321). These additions improve transparency regarding the limitations of cross-diet comparisons.

Results:

We continue to have concerns about the alignment between results presented in some figures and results reported in the text. For example, the Simpson's diversity

BIOLOGY CENTRE CAS

address: Branišovská 1160/31, 370 05 České Budějovice, Czech Republic

IBAN: CZ39 0300 0000 0006 0077 3445 | SWIFT CODE: CEKOCZPP | VAT No.: CZ60077344

phone: +420 387 771 111 (telephone exchange) | www.bc.cas.cz | e-mail: bc@bc.cas.cz

result presented on lines 212-214 is not supported by Figure 6B, and the figure caption does not contain any additional information to clarify this discrepancy.

A: We thank the reviewer for carefully checking the alignment between the text and figures. We identified an error in the original script that resulted in an incorrect x-axis scale in Figure 6B. This has now been corrected.

We have clarified in both the text and the figure legend that alpha diversity was calculated using Simpson's index ($1 - D$), where values approaching 1 indicate higher diversity. The revised Figure 6B now displays the correct 0–1 scale, and the corresponding sections in the Results (lines 216-217), Discussion (line 382), and Methods (line 625) have been updated. The corrected figure and text consistently show higher microbial diversity in rats fed the Accessible Fiber diet.

Furthermore, while the immune data we requested is now discussed in the main text, based on the figure these results are not being represented accurately. For example, Figure 9 shows significant differences between colonized and non-colonized rats for IL-1 β , IFN γ , and IL4, but the text states that there is no differences in any of these cytokines between the groups and goes on to interpret the results based on a seeming erroneous conclusion (lines 253-259). It is not clear to us whether the statistical results/figures are incorrectly reported, whether the authors misinterpreted the statistics in the text, or some other issue. However, this is a major issue that must be addressed.

A: We thank the reviewer for identifying this issue. We apologize for the error in the originally reported textual description. After re-examining the dataset, we confirmed that the underlying data and statistical analyses were correct, but the corresponding text misrepresented the outcomes. This has now been fully corrected. The Results section (lines 256–262) and Discussion section (lines 426–440) have been rewritten to accurately reflect the significant differences shown in Figure 9. We sincerely appreciate the reviewer's careful attention and patience.

Minor comments

Line 66: Helminths have not “largely vanished from modern populations”. Approximately 2 billion people are infected with helminths globally. If this statement is meant to refer only to Western nations, the authors should state this clearly.

A: Has been corrected (line 66).

Line 122: A quantitative definition/source for “normal length” in *H. diminuta* is needed here.

A: Has been incorporated (line 127).

REVIEWER #3 (REMARKS TO THE AUTHOR):

BIOLOGY CENTRE CAS

address: Branišovská 1160/31, 370 05 České Budějovice, Czech Republic

IBAN: CZ39 0300 0000 0006 0077 3445 | SWIFT CODE: CEKOCZPP | VAT No.: CZ60077344

phone: +420 387 771 111 (telephone exchange) | www.bc.cas.cz | e-mail: bc@bc.cas.cz

The authors have addressed my previous minor concerns, and their major revision has significantly clarified the scope of the manuscript. The reorganization of the manuscript, including the revised introduction and the separation of the discussion section, has also improved overall clarity. I have no additional general comments beyond what I stated in my initial review.

A: We thank the reviewer for the positive feedback. This is greatly appreciated.

Minor comments:

* In the revised and relocated Table 1, the main components of the Western diet are not fully represented. Only butter fat and cholesterol are listed, whereas the original supplementary table included several additional ingredients. This appears to be a mistake.

A: We apologize for this oversight. The missing components have now been added to the revised Table 1.

* The new discussion is helpful and more clearly framed, but it is quite long. I recommend trimming redundant or lower-priority content to improve focus and readability.

A: We thank the reviewer for the recommendation. We have trimmed and refined the discussion to reduce redundancies (see tracked changes). However, more substantial cuts would compromise the interpretation of complex, multi-layered datasets, so we propose retaining the current streamlined version. We appreciate the reviewer's understanding.